

# Precipitation reconstructions for Paris based on the observations of Louis Morin, 1665–1713 CE

Thomas Pliemon[1], Ulrich Foelsche[1,2], Christian Rohr[3,4], and Christian Pfister[3]

[1]Institute for Geophysics, Astrophysics and Meteorology/Institute of Physics (IGAM/IP), University of Graz, Graz, Austria
[2]Wegener Center for Climate and Global Change (WEGC), University of Graz, Graz, Austria
[3]Oeschger Centre for Climate Change Research, University of Bern, Bern, Switzerland
[4]Institute of History, Section of Economic, Social Environmental History (WSU), University of Bern, Bern, Switzerland

**Correspondence:** Thomas Pliemon (thomas.pliemon@uni-graz.at)

**Abstract.** This paper presents a precipitation reconstruction that is based on the continuous observations by Louis Morin in Paris from 1665–1713. Morin usually recorded precipitation intensity and duration, when it snowed/rained, three times each day (sometimes up to six times). The consistency of his observations can be calculated from his other measurements and observations (e.g. temperature, cloud cover), where at least one entry of his different measurements and observations for 98.7 % of all days is noted. To convert these observations to common units, we calibrated them with a multiplicative interacting model using Philippe and Gabriele-Philippe de la Hire's instrumental measurements from Paris. The two series of measurements by de la Hire (father and son) and observations by Morin overlap from 1688–1713. To test the quality of the reconstruction, we analyzed it with the de la Hire's measurements, proxy data, an internal analysis of Morin's measurements of different climate variables, and modern data. Thus, we assess the reliability of the precipitation reconstructions based on Morin's data as follows. We have moderate confidence regarding the exact quantities of daily, seasonal, and annual precipitation totals. We have low confidence regarding exceptionally high precipitation amounts, but we have high confidence in the indices of an impact analysis (i.e., dry days, wet days, consecutive dry days, consecutive wet days), in monthly frequencies of rainfall, and in interannual, interseasonal, and interdecadal variability. Rainy seasons with precipitation totals greater than 250 mm occurred in MAM 1682, JJA 1682, SON 1687, JJA 1697 and JJA 1703. Furthermore, compared to other DJF seasons, the winter of 1666/67 stands out with a precipitation total of 248 mm. Dry seasons with precipitation totals less than 60 mm occurred in SON 1669, DJF 1690/91 and DJF 1693/94. An impact analysis shows no abnormalities regarding consecutive dry days or wet days in MAM. In JJA a longer dry period of 31 days appeared in 1686 and a dry period of 69 days appeared in DJF 1671/72.



## 1 Introduction

Precipitation and temperature are the most important climatic elements that affect human economies and terrestrial ecosystems.
The interest in climatic data from previous centuries results (on the one hand) from the simple interest to describe the climate
in the past and (on the other hand and more importantly) to analyze climate variability and extremes in the context of climate
change. Given that precipitation is far more spatially variable than temperature, a higher density of stations and measurements
is needed to assess historical precipitation patterns. The history of rain gauge measurements is extensive. The first known
references date back to India in the fourth century BCE and Palestine in the second century BCE (Strangeways, 2010). This
instrument was practically unknown in Europe until Benedetto Castelli "invented" the rain gauge in 1639 (Camuffo, 2018;
Camuffo et al., 2020). However, most precipitation stations were only setup in the twentieth century. Even in Europe and
the United States only a few longer instrumental series were made before 1850 (e.g. Auer et al., 2001; Wigley et al., 1984;
Camuffo, 1984; Slonosky, 2002; Murphy et al., 2018; Brönnimann et al., 2019; Camuffo et al., 2019, 2020). This shortage
may be connected to the fact that rain-gauges, unlike thermometers and barometers, were not standardized and manufactured
in large quantities in the eighteenth and early-nineteenth centuries (Gimmi et al., 2007).

In consequence of this scarcity of continuous measurements of precipitation in the early instrumental period, the EU IM-
PROVE (Improved Understanding of Past Climatic Variability from Early Daily European Instrumental Sources) project aimed
to improve our knowledge of past climatic variability from early daily instrumental sources but only focused on air-pressure and
air-temperature (Gimmi et al., 2007), it did not include early instrumental series of precipitation (Camuffo and Jones, 2002).
Little is known about precipitation patterns in the "Little Ice Age" or even in the period prior to a substantial anthropogenic
forcing in the twentieth century (Bradley and Jones, 1993; Lean et al., 1995; Mann et al., 1998). Thus, to achieve a higher
density of precipitation data, spatially as well as temporally, more methods were developed to reconstruct past variations of
precipitation. A general distinction is made between archives of nature (nature-generated data) and archives of societies (an-
thropogenic data) (White et al., 2018). For instance, apart from instrumental measurements, the former include deondroclimatic
or lake sediments data (e.g. Rinne et al., 2013; Labuhn et al., 2016a). The latter include data of weather chronicles, weather
diaries, ship logbooks, weather reports, agricultural production, snow cover or floods (e.g. Pfister et al., 1999; Wheeler and
Suarez-Dominguez, 2006; Rohr, 2006; Glaser, 2008; Wheeler et al., 2010; Rohr, 2013; Dobrovolný et al., 2015; Brázdil et al.,
2016; Brázdil et al., 2018).

Louis Morin from Paris recorded daily, per eye-witnessed observations, quantitative values for precipitation amount and
intensity. Consequently, his records count as narrative observations or, more accurately, as a weather diary. Therefore, Louis
Morin's narrative data of precipitation are of great interest because he consistently recorded precipitation and other meteoro-
logical variables three times a day from 1665 to 1713 (Legrand and Le Goff, 1987; Pfister and Bareiss, 1994; Pliemon et al.,
2022). Some years (1688–1713, except 1691, 1692, 1697 and 1698) overlap with the instrumental measurements by Philippe
de la Hire and his son Gabriel-Philippe in Paris, but these are available only at monthly resolution (Slonosky, 2002). To our
knowledge, only Pfister and Bareiss (1994) dealt with Morin's precipitation notes and showed snow- and rainfall frequen-



cies from 1675 to 1713. Furthermore, Pfister and Bareiss (1994) supposed that a calibration of precipitation totals leads to non-significant results on a monthly basis.

The objective of this paper is to provide reconstructions of precipitation using Morin's eye-witnessed observations. This is done by discussing the most appropriate transfer function and analyzing these weaknesses of the reconstruction. With the awareness of the weaknesses of the reconstruction, individual indices of precipitation are discussed and thus the climate of Paris in terms of precipitation of the late-seventeenth century and early-eighteenth century is presented. Section 2 introduces the observer Morin and his meteorological journal. In Section 3, we introduce the transformation of Morin's observations into common units (i.e., the calibration methods). In Section 4, we discuss the reliability of the reconstruction by comparing them with with the measurements of Philippe and Gabriel-Philippe de la Hire, proxy data, an internal analysis of Morin's measurements of different climate variables, and modern data. Furthermore, we present different time series and conduct an impact analysis in Section 4. The last section sums up the results.



## 2   Data

### 2.1   The observer Louis Morin and his meteorological journal

Louis Morin lived from 11 July 1635 to 1 March 1715 in Paris. The majority of Morin's measurements and observations (e.g.,
temperature, pressure, direction of the movement of the clouds, etc.) were performed three times a day. A detailed explanation
of his measurements and observations, as well as his biography, has been presented in previous studies (Legrand and Le Goff,
1987; Pfister and Bareiss, 1994; Pliemon et al., 2022). Because Morin had a fixed daily routine, it is suggested that these
measurements and observations were done at around 6 am, between 11 am and 2 pm, and between 6 and 7 pm (Pfister and
Bareiss, 1994). Cornes et al. (2012) estimated the observation times at 6 am, 3 pm, and 7 pm. Further evidence of the times
of the measurements was provided by Pliemon et al. (2022) using a statistical analysis, which suggested measurement times
at 6 am to 8 am, 3 pm to 5 pm, and 6 pm to 8 pm. However, rainfall was noted to sometimes be in aligned with the other
measurements (see Fig. 1; 6 August 1702), and sometimes in-between (see Fig. 1; 7 August 1702). A further consequence
of his fixed daily routine is that his measurements show just three gaps with more than 10 consecutive missing days. These
periods are 7 June 1666 to 6 September 1666, 24 February 1668 to 18 March 1668 and 1 December 1673 to 12 December 1673.
However, in the latter two time periods he noted at least sometimes non-instrumental observations, such as the direction of the
movement of the clouds and precipitation. This reflects the consistent record of his meteorological journal, which shows at
least one entry for 98.7 % of all days.

Morin's location of residence changed several times within Paris, which is known from the rare notes of Louis Morin
(Legrand and Le Goff, 1987). Until October 1685, he lived on the Quinquempoix Street. Then, until June 1688 he lived in
the Hotel Rohan-Soubisse, where the National Archives are located today. Until his death in 1715, he lived in the Abbey of
Saint-Victor, which was located at the former city border next to the Seine (Legrand and Le Goff, 1987; Pfister and Bareiss,
1994; see Pliemon et al., 2022: Fig. 02 to find the localization on a city map of that time).

Morin measured and observed several meteorological variables (Legrand and Le Goff, 1987; Pfister and Bareiss, 1994;
Pliemon et al., 2022) and has recorded them in a well-structured manner from 1665 to 1709 in his meteorological journal.
After that, from December 1709 to June 1713, his measurements and observations are noted on loose paper (Pfister and Bareiss,
1994). Figure 1 shows as an example the first days of August 1702. As highlighted in Fig. 1, the precipitation observations are
entered in column 14. Before discussing the precipitation notes in more detail, we briefly introduce his other measurements
and observations. Columns 1 to 4 represent the day of the month; the day of lunar cycle; the conjunction, opposition and
other aspects of the moon and the sun; and the conjunction, opposition and other aspects of the planets, respectively. The
latter three are mostly empty. This is followed by the thermometer measurements (column 5), the hygrometer measurements
(column 6), and the barometer measurements (column 7). These three values were measured instrumentally, and—except for
the hygrometer measurements (from 1701 to 1711)–were consequently performed over the whole period. Columns 8 to 13 give
the direction of the wind (seldom noted), the strength of the wind (often noted), the direction of the movement of the clouds
(often noted), the regional origin of air (often noted), the speed of the clouds (often noted) and the cloud cover (often noted),
respectively. Finally, the last column (column 15) gives details for fog, snow, and so on.



Although rain gauges were already relatively common during and before Morin's lifetime (Strangeways, 2010), he made eye-witnessed observations of precipitation. This means that he subjectively noted the intensity and duration of precipitation. The notes consist of one letter and two numbers, where the former indicates a snow (letter "n"; in french: neige) or a rain event (letter "p"; in french: pluie). The first number denotes rain intensity (RI) and the second rain duration (RD). Both RI

and RD are quantified by numbers between 1 and 6, where 1 means low RI / short RD, respectively, whereas 6 means high RI / long RD, respectively. Furthermore, single "p"-notes represent light rainfall and single "n"-notes a light snowfall. He quite often noted just a single number. For instance, we interpret a single number of 2 as a RI of 2 and a RD of 0. To check the distribution of values for abnormalities, we plotted the total occurrences of each value in Fig. 2 (for visualization "p" entries get the number 0 and "n" entries get the number -1). Note that the y-axis has a logarithmic scale. Furthermore, the x-axis is

read in such a way that RI indicates the first number and RD the second number, thus leaving non-existing notations free. The following conspicuous points can be determined: (1) Morin strongly reduced or did not note numbers with a value of five for both RI and RD; and (2) he has a strong preference for RI = 2. The latter is more pronounced for rainfall than for snowfall and results in a lower variance of RI compared to RD. A tendency to prefer certain values was already seen when analyzing the notes of the temperature (Legrand and Le Goff, 1987).

Given that his observations were made on a subjective basis, we have also analyzed the time series of individual values. We can see that he made a small change in his notes. While notes with RD = 1 decrease, others with RD = 0 increase for notes in the period between 1679 and 1681. Furthermore, until September 1665, he only listed the values "n" and "p". He only introduced the further categorization/quantification with numerical values from October 1666. Furthermore, notes of "n" are rare up to 1680, while they increase later to upto 50 times per year (see Fig. A1).

## 2.2 Reference data

We used the modern observations of E-OBS version 26.0e (Cornes et al., 2018). We follow the WMO (World Meteorological Organization, 2017) and chose the historical base period (1961 to 1990) as 30-year reference normal. Furthermore, we use indices of the North Atlantic Oscillation (NAO; Luterbacher et al., 2001) and dendroclimatological data (Labuhn et al., 2016b). Slonosky (2002) examined instrumental precipitation measurements of Philippe and Gabriel-Philippe de la Hire. Because the

measurements performed by father and son de la Hire overlap with the observations by Morin, we used these data for calibration and also for comparisons. Nevertheless, we have to expect measurement errors with rain gauges. An important error source results from wind, especially when given that rain gauges used to be positioned at higher altitudes (Auer et al., 2005; Camuffo et al., 2020, 2022a). This leads to reduced precipitation totals (PTs) with stronger influence in winter due to snowfall than in the remaining seasons. Other possible influencing factors are evaporation (lower PT; Camuffo et al., 2020) and dew

(higher PT; Camuffo et al., 2019, 2020). The influence of dew and evaporation is negligible because father and son de la Hire recorded precipitation as soon as it fell (Slonosky, 2002). However, due to the higher elevation of the measurements sites, the influence of wind is significant. The copper box (measurement device) of the de la Hires was based in an uncompleted tower of the observatory, which had four walls with window holes but no roof (Slonosky, 2002). So, Slonosky (2002) updated the data file in 2019, where especially the values of winter months have been adjusted upwards.



## 3 Methods

### 3.1 Assumptions and data modification

To get common units, we applied a transfer function to Morin's precipitation data. Before doing so, we made two modifications to the data. First, notes without a RD specification are set with a RD = 1. The reason for this modification of the raw data has already been mentioned, namely that Morin's attribution of rainfall events without a specification of time in earlier years was very likely noted as RD = 1 in later years. The second is more of an assumption and concerns the notes with "n" and "p", which are interpreted as short and light snow- and rainfall events. In previous studies different thresholds of the precipitation amounts are given, which can be recognized without instrumentation: 0.1 mm (Glaser, 2008), 0.2 mm (Brumme, 1978), and 0.3 mm (Pfister and Bareiss, 1994; Gimmi et al., 2007). Glaser (2008) suggests that a precipitation amount of 0.1 mm is detectable during the day due to wet roads and roofs, and reduced precipitation amounts compared to instrumental data are more likely to occur due to unnoticed precipitation amounts at night. Based on our data, we are unable to analyze the reasons for inhomogeneities and decided to use a minimum perceptible precipitation amount of 0.3 mm. In total, Morin noted about 15 % of all values with either "n" or "p". Derived from previous analyses (Gimmi et al., 2007, Tab. 3), we set the upper limit at 0.7 mm. Thus, we assume that the precipitation amount of these precipitation events lies between 0.3 mm and 0.7 mm, and therefore assign these values with the mean of 0.5 mm. We do not account for the rise of "p" values since 1680 (see Fig. A1), which may slightly overestimate years without "p" values. Because of Morin's careful record keeping, we rule out another possibility of interpretation, such as that the "n" and "p" values were merely not fully noted.

### 3.2 Calibration per transfer function

Once the assumptions are made, the measured values, which consist of RI and RD, were converted into common values (mm). To our knowledge, there is no identical historical database in the literature with measurements of RI and RD. Somewhat similar are the records of Morgagni (Camuffo et al., 2022b), who noted RI but not the rain duration. From a methodological perspective, Camuffo et al. (2022b) had daily measurements available for calibration. Due to the lack of similar studies, we applied different transfer functions $f_{(RI,RD)}$. In Equ. 1, we use the simplest form, multiplying RI and RD. In Equ. 2, a multiplicative variable $a_2$ is appended with the unit millimeter. In Equ. 3, we kept the muliplicative factor and weighted RI and RD separately by the parameters $b_3$ and $c_3$. A multiplicative interaction model is shown in Equ. 4 with the parameters $a_4$, $b_4$, $c_4$ and $d_4$.

$$PT_1 = RI \cdot RD, \tag{1}$$

$$PT_2 = a_2 \cdot RI \cdot RD, \tag{2}$$

$$PT_3 = a_3 RI^{b_3} \cdot RD^{c_3}, \tag{3}$$





$$PT_4 = a_4 + b_4 \cdot RI + c_4 \cdot RD + d_4 \cdot RI \cdot RD, \tag{4}$$

where *PT* denotes the monthly precipitation total, *RI* the rain intensity, *RD* the rain duration, and $a_x$, $b_x$, $c_x$ and $d_x$ are
parameters. For the calibration, monthly PTs were taken from the measurements by the de la Hires (Slonosky, 2002). Their
available measurements started in June 1688 and are mostly noted throughout except for the years 1691, 1692, 1697, and 1698.
This means that about 20 monthly values for each month in the measurement series overlap between 1688 and 1713. This time
span is too short to us to separate it into a calibration time period and a validation time period. Therefore, we use this time
period between 1688 and 1713 for both calibration and validation. The de la Hire measurements are instrumental measurements
and are available in a monthly resolution, and thus this is a calibration on a monthly basis.

A clear choice of the parameters in Equ. 1-4 could not be made by the least squares method alone, because some parameter
constellations achieve approximately equally good/minimal results. Consequently, constellations of parameters have been se-
lected, which exceed the minimum root mean square error (RMSE) and mean average error (MAE) by less than one percent
(see Tab. A1, but note that due to the high number of possibilities we only show errors less than three per mill for the warmer
period). Of the values that meet these conditions, the choice was made according to either the strongest weighting of the mul-
tiplicative term (RI · RD) or the highest degree of retention of the original values of RI and RD. The heavy weighting on a RI
value of 2 leads Morin to underestimate summer values. We address this problem by calculating the parameters for the summer
months (May–September) and the remaining months separately. A different calibration for each month (Jan, Feb,...) does not
seem to be reasonable due to the small number of individual months possible for calibration. As an example, all parameters
that fulfill the conditions for Equ. 4 are listed in Tab. A1. For Equ. 3 we chose the parameters $a_2 = 0.6$ (October to April)
and $a_2 = 0.9$ (May to September). For Equ. 3 we chose the parameters $a_3 = 0.7, b_3 = 1.2, c_3 = 0.7$ (October to April) and
$a_3 = 0.7, b_3 = 1.7, c_3 = 0.7$ (May to September). For Equ. 4 we chose the parameters $a_4 = 0.0, b_4 = 0.2, c_4 = 0.0, d_4 = 0.5$
(October to April) and $a_4 = 0.0, b_4 = 0.6, c_4 = 0.0, d_4 = 0.6$ (May to September).

### 3.3    Calibration per rainfall frequency

For the months Jan. 1665 to Sept. 1665, only precipitation events are documented and no rain intensity or rain duration are
documented. Thus, we calculated the PT for these months using a calibration (linear regression) with the rain frequency. That
is, a given rainfall frequency receives a certain PT.

### 4    Results

To find the strengths and weaknesses of the data, in the following two subsection we compare Morin's observations with both
contemporary and modern precipitation measurements, and with proxy data. These comparisons also serve as a basis for a
discussion of the various calibration methods and selected one of Equ. 1–4. The last subsection presents the time series of
Morin's precipitation observations and an impact analysis.



### 4.1 Calibration/validation and choice of the calibration method

To compare the de la Hire data with calibrated data of Morin according to Equ. 1–4, we created a scatterplot of each in

Fig. 3. Thereby, method 2 strongly underestimates the measurements of father and son de la Hire, and methods 1 and 2 show relatively high scatter. Method 3 and 4 provide a good result, showing a good correlation and only slightly underestimating the de la Hire data. In terms of correlation (Pearson), method 1 and method 2 reveal 0.59 and 0.69 (1688–1713), respecitvely, whereas method 3 and method 4 show, rounded to the second digit, a correlation of 0.73. However, more analysis and reasons for the underestimation of the de la Hire data are needed to commit to a calibration method.

The monthly means for the calibration period of 1688 to 1713 (excluding missing data; see Sect. 3) are shown in Figure 4. The correlation coefficient is noted in each panel. If the calibration period is not split into summer and winter periods, then the monthly means show an underestimation of the summer months and an overestimation of the winter months for all calibration methods. Even when the periods are calibrated separately, the summer months (especially May and July) show larger deviations (see Fig. 4). It is also easy to see that calibration methods 1 and 2 do not provide satisfactory results. However, calibration

methods 3 and 4 show a good agreement, with slightly lower precipitation amounts in summer. Using this graph, we settle on calibration method 4, which will be used to process the following evaluation in this paper. The results of this method and the calibration method 3 are similar. However, 8.8 % values of RI ($RI = 1$) and 35.9 % values of RD ($RD = 1$) do not affect the result of method 3 due to the base one. Even though Equ. 3 achieves slightly better results in terms of RMSE and MAE, Equ. 4 seems reasonable to us because of the reason given earlier and the fact that it is the mathematically simpler regression

formula.

Precipitation totals based on eye-witnessed observations tend to underestimate heavy rainfall (Camuffo et al., 2022b). Thus, we looked at whether the data reflect extreme events. Philippe and Gabriel-Philippe de la Hire, Bonamy, and Deparcieux highlighted the years 1658, 1711, and 1740, in which flooding occurred in Paris (Slonosky et al., 2020). February 1711 falls in our time period, and the flood level of this year is also recorded on buildings (e.g. Paris, 29 place Maubert, 5e arr.). However,

this flooding was not only due to heavy rain but was a combination of snowmelt and rain (Slonosky et al., 2020). As expected, the de la Hire data reflect this exceptional year better (high PT in February) than Morin's data (see Fig. 5). In this figure, the comparisons of the PT between Morin and the de la Hires are plotted seasonally and annually as a time series. Furthermore, we find an annual Pearson correlation of 0.72.

Another possibility of validation is a comparison with proxy data. The latewood tree ring isotope $\delta^{18}$O (Labuhn et al., 2016a,

Fontainebleau) significantly correlates with the growing season maximum temperatures, as well as with precipitation. The correlation is more significant for temperature than for precipitation. However, Etien et al. (2009) compared summer (JJA) and annual precipitation amounts with $\delta^{18}$O for Fontainebleau. They showed that these variables are significantly anti-correlated from 1900 to 1950 (R = -0.50 and p = 0.0002 for $P_{JJA}$, R = -0.59 and p = $9 \times 10^{-6}$ for $P_{ann}$) but the anti-correlation is weaker since 1950 (R = -0.38 and p = 0.006 for $P_{JJA}$, R = -0.32 and p = 0.02 for $P_{ann}$). Our correlation (Spearman) analysis for

$PT_{JJA}$ revealed a correlation of r = -0.30 p = 0.04 and for $PT_{ann}$ a correlation of r = -0.42 and p = 0.003. Thus, the correlation



with our data reveals a weaker anticorrelation and both are significant at the 0.05 level. The time series of both $\delta^{18}O$ and the monthly average of PT of each year are plotted in Fig. 6.

A comparison with the NAO is difficult because NAO data itself is based on reconstructions for the time period of interest. The further in the past, the less predictors and the larger the error (Pauling et al., 2006). In theory, in contrast to temperature,
there should be no correlation in Paris between precipitation and the NAO index (Cleary et al., 2017; Müller-Plath et al., 2022). This means that other climatic drivers are responsible for the precipitation pattern over France. Morin's data show in DJF $r = 0.01, p = 0.97$, and in JJA $r = 0.21, p = 0.16$, and so the data show what is expected.

A further, but weaker, possibility of validation is to intercompare Morin's various meteorological variables. Looking at selected meteorological variables influenced by precipitation, differences should be noticeable. We compared three different
parameters (see Tab. 1). Given that precipitation is sometimes not in line with the usual three measurements per day (Pliemon et al., 2022), we made the comparison based on daily mean values. Total cloud cover (TCC) reflects the expected relationship: the higher the PT, the higher the TCC. In detail, the TCC increases from 3.6 for prec = 0 to 7.5 for prec >= 15 mm. Diurnal temperature range (DTR) is also consistent: the higher the PT, the smaller the value for DTR. In detail, the DTR decreases from 7.6 °C for prec = 0 to 4.9 °C for prec >= 15 mm. Humidity measurements were made instrumentally by Morin. However,
we do not know which instrument he used and there is no information about the metadata or the implementation. Thus, we made the comparison with the units noted by Morin. Here, a positive value means humid air and a negative value means dry air. Again, the calculated values are plausible: the higher the PT, the more humid the air. Nevertheless, the values for prec >= 10 mm and prec >= 15 mm show smaller values.

| | Prec = 0 mm | Prec < 1 mm | Prec >= 1 mm | Prec >= 5 mm | Prec >= 10 mm | Prec >= 15 mm |
|---|---|---|---|---|---|---|
| TCC (octas) | 3.6 | 5.2 | 5.8 | 6.5 | 7.0 | 7.5 |
| DTR (°C) | 7.6 | 6.8 | 6.3 | 5.9 | 5.2 | 4.8 |
| Hum (MU) | 0.9 | 1.5 | 1.8 | 1.8 | 1.5 | 1.4 |

**Table 1.** Various meteorological values measured and observed by Morin as a function of precipitation. Total cloud cover (TCC) in octas (without fog days), diurnal temperature range (DTR) in °C, and humidity in Morin's noted unit.

## 4.2   Comparison/validation with modern measurements

We also compared the results of the reconstruction with modern data, primarily for the plausibility check. However, if we felt confident enough to make stronger statements about differences, then this is explicitly noted. As a modern data set, we used the E-OBS data for the period 1961–1990 (Cornes et al., 2018). The daily resolution of the E-OBS data allows us to compare not only the monthly precipitation totals but also the monthly frequency of wet days (see Fig. 7). Although Morin was known for consistently making his measurements and observations with only a few misses (Legrand and Le Goff, 1987;
Pliemon et al., 2022), looking at the frequency of precipitation serves as further validation of the reliability of his data. Over the year, the precipitation totals in Paris are approximately evenly distributed. This is true for both the E-OBS data and Morin's



observations, but the annual cycle is more pronounced for Morin's observations. The monthly frequency of wet days (PT >= 1mm) shows an almost uniform distribution and ranges from about 8 to 12 wet days on average per month. Interestingly, in contrast to Morin's observations, the E-OBS data indicate a slightly stronger annual variation with lower values in the summer

months. However, the comparison of monthly frequencies shows no clear differences, and thus Morin's observations can be considered consistent in this regard. Similarly, the annual pattern of monthly precipitation totals, which is dependent of the calibration method, matches that of the comparison period to a high degree. Nonetheless, our model underestimates the winter months. The reasons for this could be that the calibration data are subject to higher measurement inaccuracies in the winter months (see Sect. 2.2) or that Morin's records underestimate snowfall because the majority of the records correspond to only

0.5 mm (see Fig. 2).

The frequency of daily precipitation totals (a) and the frequency of consecutive wet days (b) are plotted in Fig. 8. In theory, the frequency of the daily precipitation totals follows a gamma distribution (e.g. Thom, 1958; Martinez-Villalobos and Neelin, 2019). This is well-satisfied for the E-OBS data and Morin's observations also roughly follow this distribution. Only the frequencies of light precipitation show stronger deviations. This is due to the discrete nature of his observations. This effect

varies strongly with the choice of the calibration method (see section 3). In particular, the peak of the frequency for precipitation of the interval (1-2] mm is underestimated while the interval (0-1] mm is overestimated. However, many values are equal to 1 mm due to the calibration method, and thus we feel that an impact analysis is possible. Furthermore, we can see that heavy rainfall events are underestimated by Morin's observation method. The underestimation of intense precipitation becomes better apparent when looking at the percentiles (Tab. 2). The higher the percentiles (especially from the 90 % decile), the

more the values diverge between Morin's calibrated observations and the E-OBS data. Morin's calibrated precipitation totals underestimate heavy precipitation, as seen previously. Especially from the 97 % percentile on, the values diverge and already show a difference of 1.9 mm and more, whereas up to the 80 % decile there is good agreement. In Fig. 8(b), we see a very good agreement in the frequency of consecutive wet days between the E-OBS data and Morin's observations. Given that this comparison has a weaker dependence on the choice of the calibration method, we cautiously hypothesize that the studied time

period indeed shows a higher frequency of single events but a lower frequency of longer rain periods.

To summarize, following the analyses of the previous and this section, we assess the reliability of the precipitation reconstructions based on Morin's data as follows. We have low confidence regarding exceptionally high precipitation amounts. We have moderate confidence regarding daily, seasonal, and annual precipitation totals. We have high confidence for an impact analysis (dry days, wet days, consecutive dry days, and consecutive wet days), in monthly frequencies of rainfall, and in inter-

annual/interdecadal variability. Thus, the last section will present time series of precipitation and will give an impact analysis.

### 4.3   Reconstructed time series and impact analysis

First, we examine monthly variability. Thus, we present monthly precipitation anomalies and monthly precipitation frequency anomalies in Fig. 9 with respect to the monthly mean of the whole observation period. In addition, the blue shaded areas show the 11-month running mean (see Tab. A2/A3 for all absolute values of monthly and annual PT). No observations are recorded

for July 1666 and August 1666, and consequently we set the precipitation total anomaly and precipitation frequency anomaly to



| | Morin (mm) | E-OBS (mm) |
|---|---|---|
| P10 | 0.5 | 0.9 |
| P20 | 1.4 | 1.4 |
| P30 | 1.5 | 1.9 |
| P40 | 2.4 | 2.5 |
| P50 | 2.8 | 3.2 |
| P60 | 3.6 | 4.1 |
| P70 | 4.8 | 5.2 |
| P80 | 6.4 | 6.9 |
| P90 | 9.0 | 9.9 |
| P95 | 12.0 | 12.8 |
| P97 | 13.6 | 15.5 |
| P98 | 15.2 | 17.4 |
| P99 | 17.2 | 19.9 |

**Table 2.** Percentiles of the calibrated results of Morin's observations compared with the E-OBS data.

zero for these months (see Sec. 2.1). Furthermore, because only the precipitation event is noted for the months of January 1665 to September 1665, precipitation totals of those months are calculated as a function of the precipitation frequency. The highest monthly precipitation totals are noted in September 1687 (164.7 mm), August 1697 (146.8 mm), and June 1703 (151.5 mm). Nevertheless, we claim the quantity of monthly PT only with medium confidence. The highest numbers of 23 precipitation days per month were recorded by Morin in March 1693, June 1703, and March 1709. The 11-month running mean shows relatively high variability for both rainfall amounts and frequency, with moderately pronounced wet and dry periods. Basically, it can be said that no exceptional dry or wet period can be detected during Morin's observation period. Regarding the precipitation frequency anomalies in Fig. 9 (b), there is a predominance of negative anomalies up to and including 1672, positive anomalies up to and including 1683, negative anomalies up to and including 1697, and positive anomalies up to and including 1710.

Second, we examine seasonal variability. To do this, we have plotted the precipitation totals for each season in Fig. 10. The red bars indicate the number of days when Morin did not make observations. With just a few exceptions, the observations are continuous. Rainy seasons with precipitation totals greater than 250 mm occurred in MAM 1682, JJA 1682, SON 1687, JJA 1697 and JJA 1703. Furthermore, with respect to the other DJF seasons, DJF 1666/67 stands out with a PT of 248 mm. Dry seasons with precipitation totals less than 60 mm occurred in SON 1669, DJF 1690/91 and DJF 1693/94. The three highest annual precipitation totals are recorded in 1682 (788.7 mm), 1697 (758.0 mm), and 1698 (754.8 mm). The three lowest annual precipitation totals are recorded in 1669 (348.0 mm), 1691 (373.4 mm), and 1694 (354.6 mm). Morin's differentiation between rain- and snowfall allows us to analyze the winters in terms of snowfall. The days of snowfall for the seasons DJF, MAM and SON are shown in Fig. A2. Thereby, each day was considered, where at least one note with "n" was made. DJF, 1669/70,



1678/79, 1694/95, and 1696/97 stand out with 27, 22, 24, and 20 days of snowfall, respectively. In MAM 1688 and 1701,
Morin noted 9 days of snowfall.

Third, we performed an impact analysis. Here, we examined the indices wet days (daily PT >= 1 mm), dry days (PT < 1 mm), consecutive wet days (CWD), and consecutive dry days (CDD). The last two indices are important because extreme values, for example in spring or summer, may have led to crop failure. With respect to the interannual variability of wet and dry days (Fig. 11 (a)), there are no conspicuous findings. The same is true for CWD, but exceptionally long CDD periods are noted in 1669 with 48 days and in 1672 with 43 days (Fig. 11 (b)). The latter dry period is even more pronounced in DJF with 69 CDD (Fig. A3). In JJA, the year 1686 stands out with 31 CDD. No exceptional dry periods are noted in MAM. In SON, longer dry periods occured in 1669 (43 days) and 1691 (37 days).

Finally, Tab. 3 shows the decadal variability for different indices. Here we see a minimum for the annual PT in the 1660s and a maximum in the 1680s. In DJF, the maximum appears in the 1700s and the minimum appears in the 1670s. The JJA-season shows the highest mean PT in the 1700s and the lowest is in the 1660s. Furthermore, the 1660s show a high value of CDD, and the 1700s show a low value of CDD. In terms of CWD, the 1670s show the highest value with 8.7 days and the 1690s the lowest value with 6.9 days.

|  | 1665–1670 | 1671–1680 | 1681–1690 | 1691–1700 | 1701–1710 |
|---|---|---|---|---|---|
| PT (mm) | 500.2 | 577.3 | 614.6 | 597.1 | 603.5 |
| PT DJF (mm) | 129.3 | 105.7 | 118.5 | 124.0 | 141.6 |
| PT MAM (mm) | 115.2 | 162.0 | 150.3 | 160.0 | 138.6 |
| PT JJA (mm) | 98.9 | 160.4 | 173.4 | 168.0 | 184.5 |
| PT SON (mm) | 140.3 | 149.2 | 172.3 | 145.1 | 138.8 |
| Dry days (d) | 247.2 | 232.1 | 249.0 | 251.0 | 240.0 |
| Wet days (d) | 107.7 | 133.2 | 116.2 | 114.2 | 125.2 |
| CDD (d) | 27.2 | 23.2 | 22.9 | 21.0 | 18.6 |
| CWD (d) | 7.2 | 8.7 | 7.5 | 6.9 | 7.3 |

**Table 3.** Seasonal and yearly means of different indices per decade: yearly means of precipitation totals (PT), decadal means of precipitation totals (PT DJF, PT MAM, PT JJA, and PT SON), yearly means of dry days, yearly means of wet days, yearly means of consecutive dry days (CDD), and yearly means of consecutive wet days (CWD).



## 5 Conclusions

Louis Morin had a strict daily lifecycle and was conscientious. This is reflected in his careful observations of rainfall (at least
one entry of his different measurements for 98.7 % of all days), where he continuously recorded both intensity (RI) and duration
of precipitation (RD) from October 1665 to July 1713 in Paris. Due to the subjective nature of eye-witnessed observations, the
original entries were checked for homogeneity. A comparison with modern data suggests that Morin underestimated summer
months. This is the reason why we separated the calibration into two periods (May–September and October–April). Further-
more, when looking at the time series of entries in the journal, subjectively different assessments were determined over time,
which were taken into account for the calibration.

For the calibration, we compared the results of four different transfer functions (functions of RI and RD). Morin's obser-
vations overlap with instrumental measurements of the de la Hires (Slonosky, 2002) available on monthly basis from 1688
to 1713. Based on our analyses and the method of least squares error, we chose a multiplicative interaction model as transfer
function. To test the quality of the reconstruction, we analyzed it with the measurements of father and son de la Hire, proxy
data, an internal analysis of Morin's measurements and observations of different climate variables, and modern data. Thus, we
assessed the reliability of the precipitation reconstructions based on Morin's data as follows. We have low confidence regarding
exceptionally high precipitation amounts. We have moderate confidence regarding the exact quantity of daily, seasonal, and
annual precipitation totals. We have high confidence in the indices of an impact analysis (dry days, wet days, consecutive
dry days, and consecutive wet days), in monthly frequencies of rainfall, and in interannual, interseasonal, and interdecadal
variability.

The agreement of the monthly rain frequency with modern data shows that Morin documented the rain events well. Looking
at the time series of precipitation totals, there are no exceptionally strong extremes for maximum and minimum precipitation.
Rainy seasons with precipitation totals greater than 250 mm occurred in MAM 1682, JJA 1682, SON 1687, JJA 1697 and
JJA 1703. Furthermore, compared to other DJF seasons, the winter 1666/67 stands out with a precipitation total of 248 mm.
Dry seasons with precipitation totals less than 60 mm occurred in SON 1669, DJF 1690/91 and DJF 1693/94. The three highest
annual precipitation totals are recorded in 1682 (788.7 mm), 1697 (758.0 mm), and 1698 (754.8 mm), the three lowest annual
precipitation totals in 1669 (348.0 mm), 1691 (373.4 mm), and 1694 (354.6 mm). The impact analysis shows 69 days without
precipitation for DJF in 1671/72. In the growing season of the plants, MAM, no abnormalities could be reconstructed, and in
JJA a longer dry period with 31 days is noted in 1686. In summary, compared to the temperature variability (e.g. winter 1708/09;
see Pliemon et al., 2022), this period is much less conspicuous in terms of precipitation.





**Figure 1.** Example of Morin's notes (Source: Institute of History / Oeschger Centre for Climate Change Research, University of Bern). The hygrometer measurements are entered in column 6 and precipitation measurements in column 14. Precipitation data were measured by Morin directly. His records consist of two numbers, one denoting rain intensity (RI) and the second rain duration (RD). Both RI and RD are noted by numbers between 0 and 6, where 0 means low RI / short RD and 6 means high RI / long RD. Furthermore, single "p"-notes represent light rainfall and single "n"-notes represent a light snowfall (see text for the remaining variables).





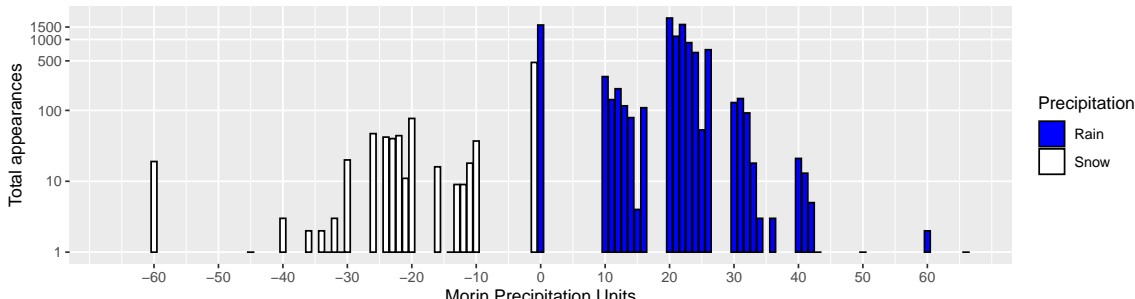

**Figure 2.** The cumulative occurrences per unit used, which was noted in Morin's precipitation records: 0 represents the note "p", meaning rain; -1 represents "n", meaning snowfall; and the two-digit numbers consist of rain intensity (first number) and rain duration (second number). Note that the y-axis has a logarithmic scale.

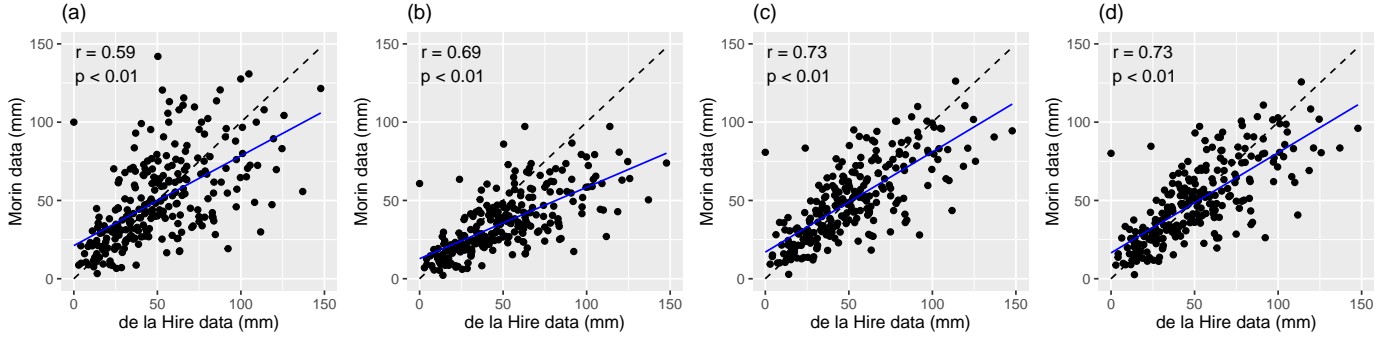

**Figure 3.** Scatterplots of the instrumental measurements of of the de la Hires (x-axis) of the different calibration methods based on Morin's data: (a) Method 1, (b) Method 2, (c) Method 3, and (d) Method 4 for the precipitation totals from 1688 to 1713. Furthermore, the Pearson correlation coefficient is noted in each panel.





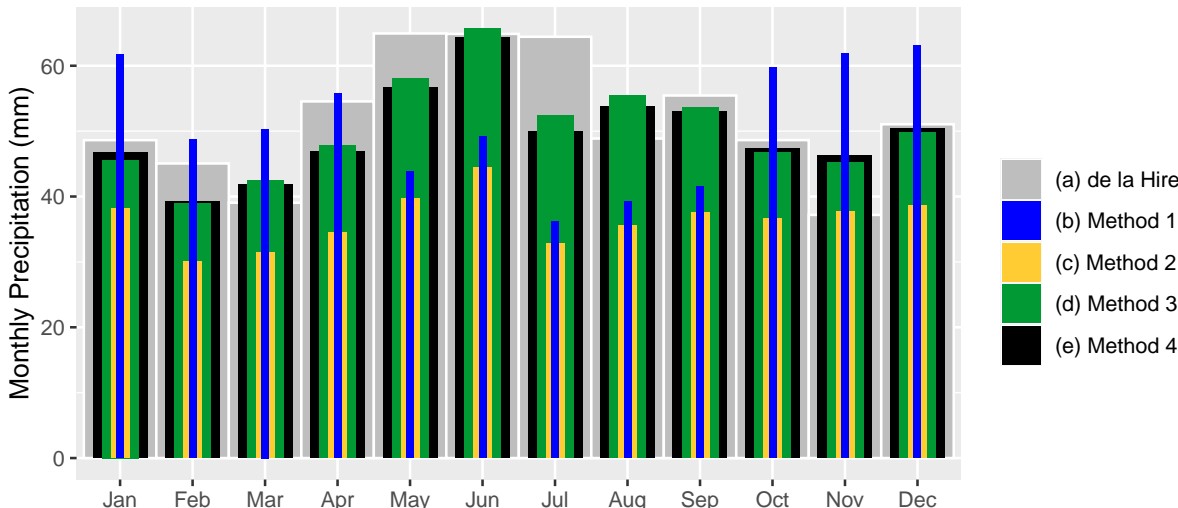

**Figure 4.** Comparison of the monthly means of the calibration period 1688–1713 (excluding 1691, 1692, 1697, and 1698) of (a) the de la Hire instrumental measurements, (b) Morin observations with calibration method 1, (c) Morin observations with calibration method 2, and (d) Morin observations with calibration method 3.




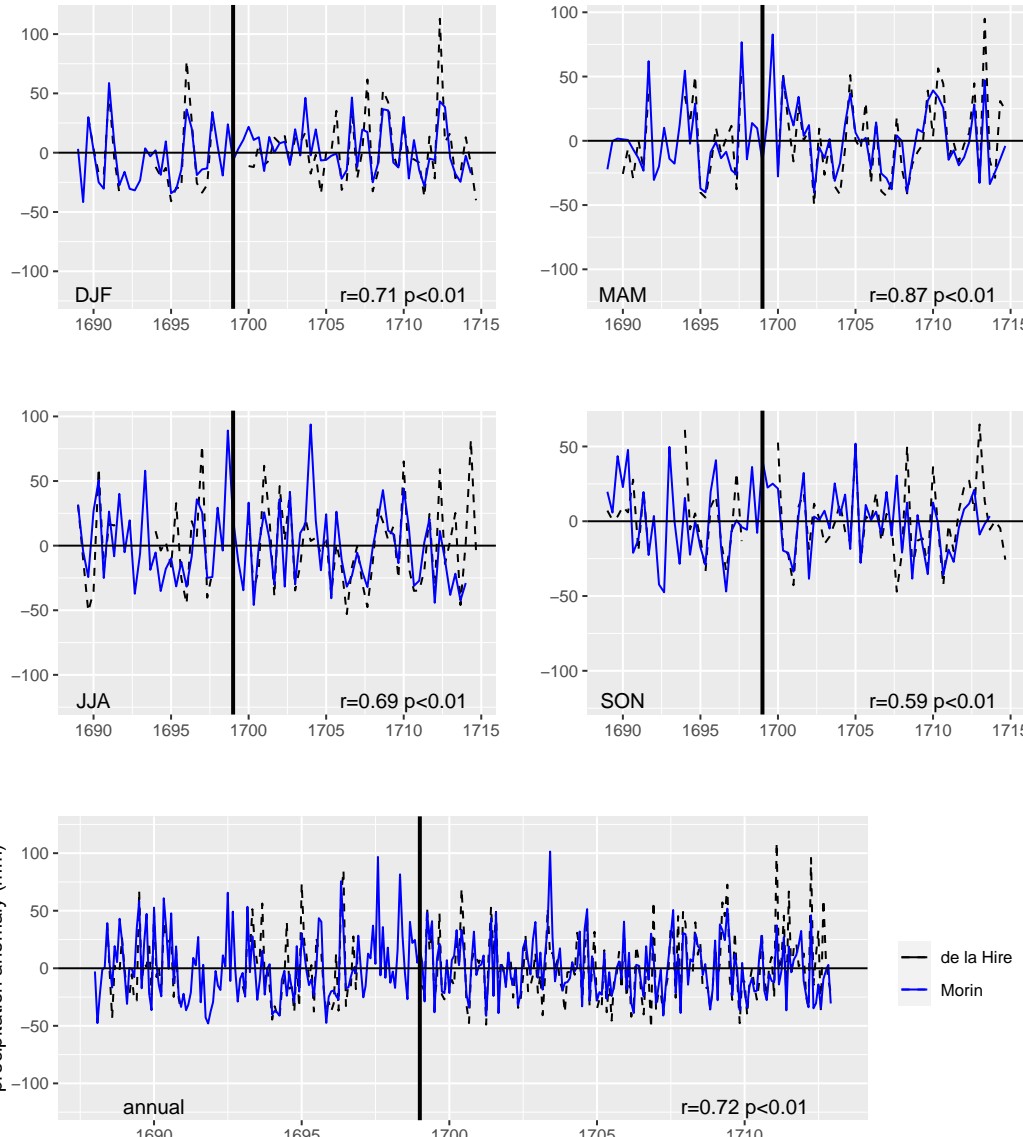

**Figure 5.** Measured (dashed line) and reconstructed (solid line) monthly anomalies of each season and annual. The vertical black line marks the transition from continuous measurements (>1698) of father and son de la Hire and non-continuous measurements. The Pearson correlation coefficient was calculated for the continuous measurement period.




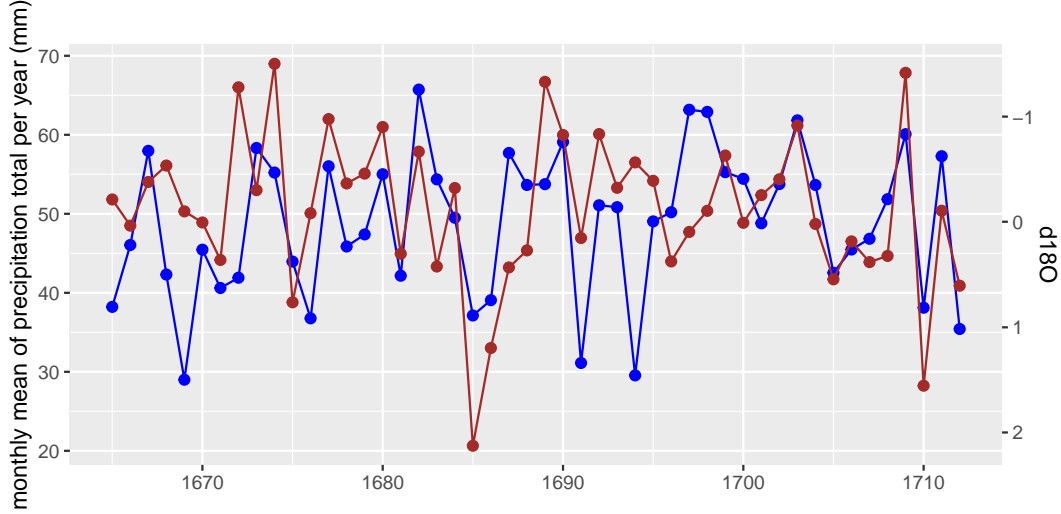

**Figure 6.** Comparison of the the monthly average of the precipitation total of each year (blue) and the $\delta^{18}$O (Labuhn et al., 2016b) (red).

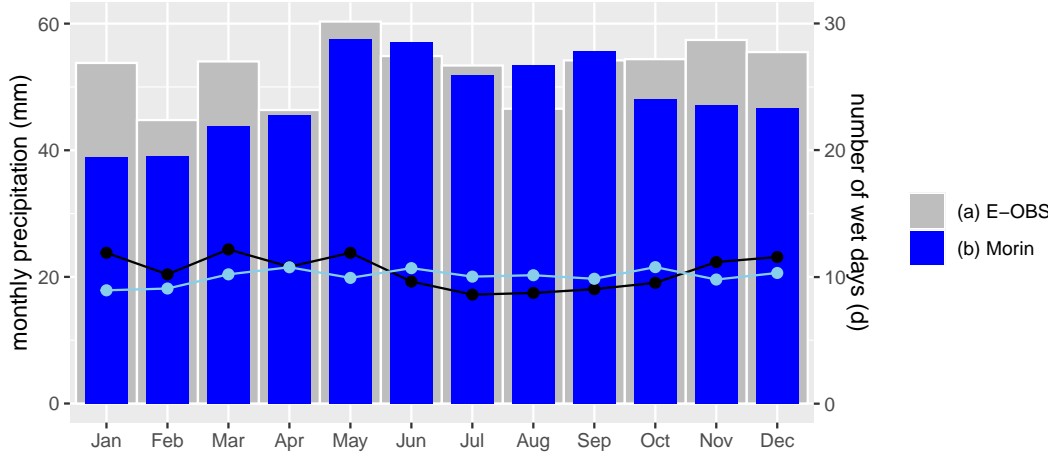

**Figure 7.** The monthly means of Morin's precipitation reconstruction (blue bars) and of the reference period (gray bars; 1961–1990). Furthermore, the blue and black line represent the number of wet days for Morin's observation and modern data (E–OBS), respectively.





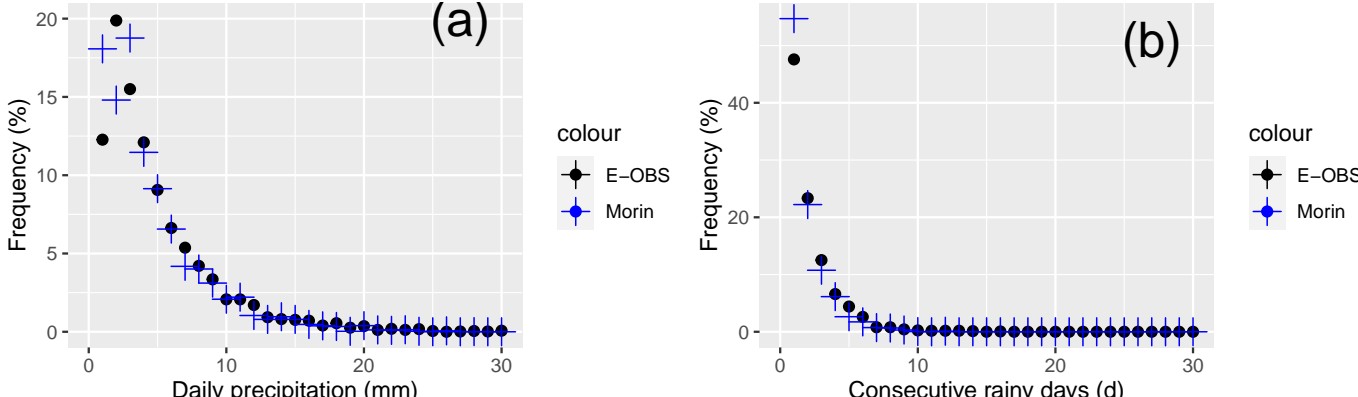

**Figure 8.** Frequency of precipitation as a function of daily precipitation of Morin's precipitation reconstruction (blue dots) and of the reference period (1961–1990). And the frequency of precipitation as a function of consecutive wet days.





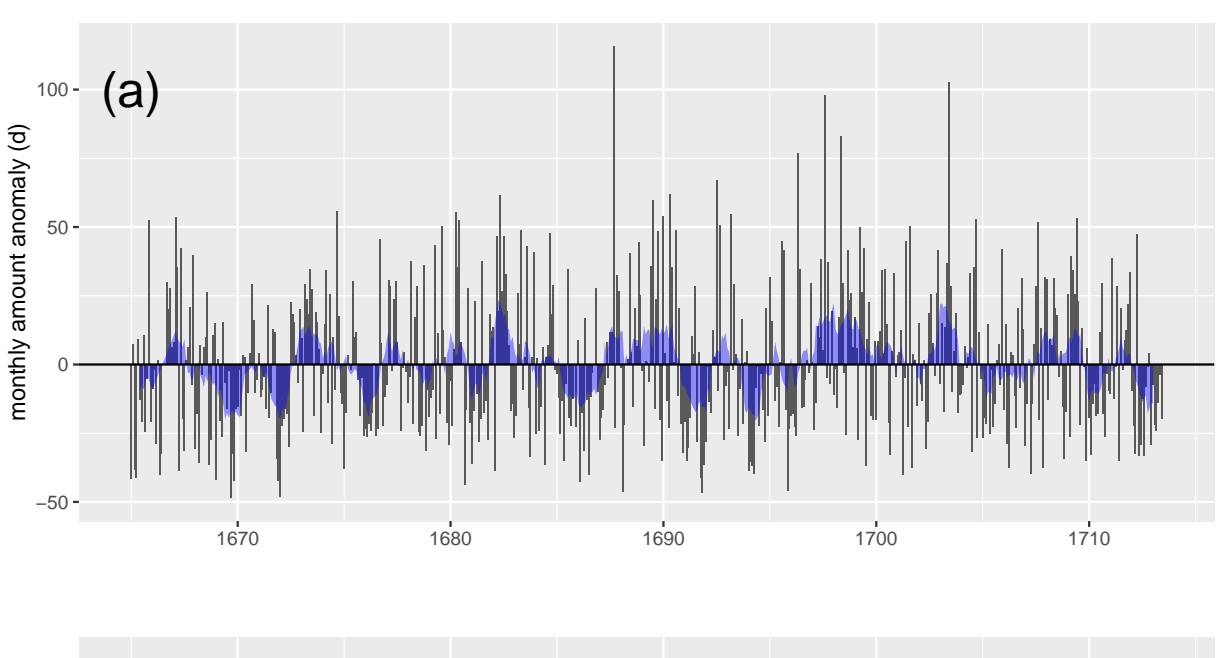

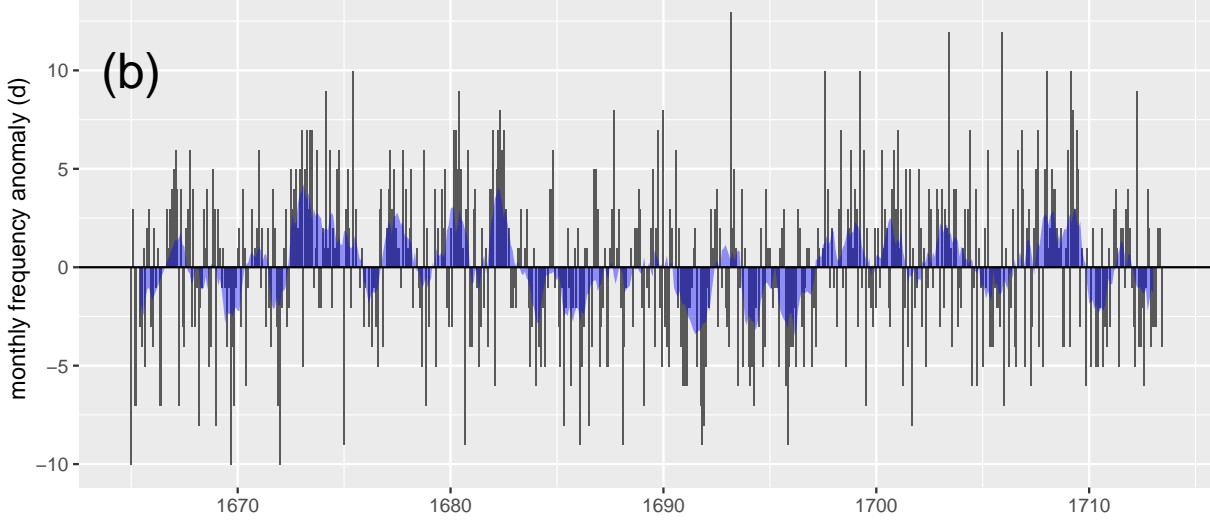

**Figure 9.** In (a), monthly precipitation anomalies are plotted from 1665–1713 with respect to the monthly mean of the whole observation period. In (b), the monthly precipitation frequency anomalies are plotted. In each case, the blue shaded curve shows the 11-month running mean.





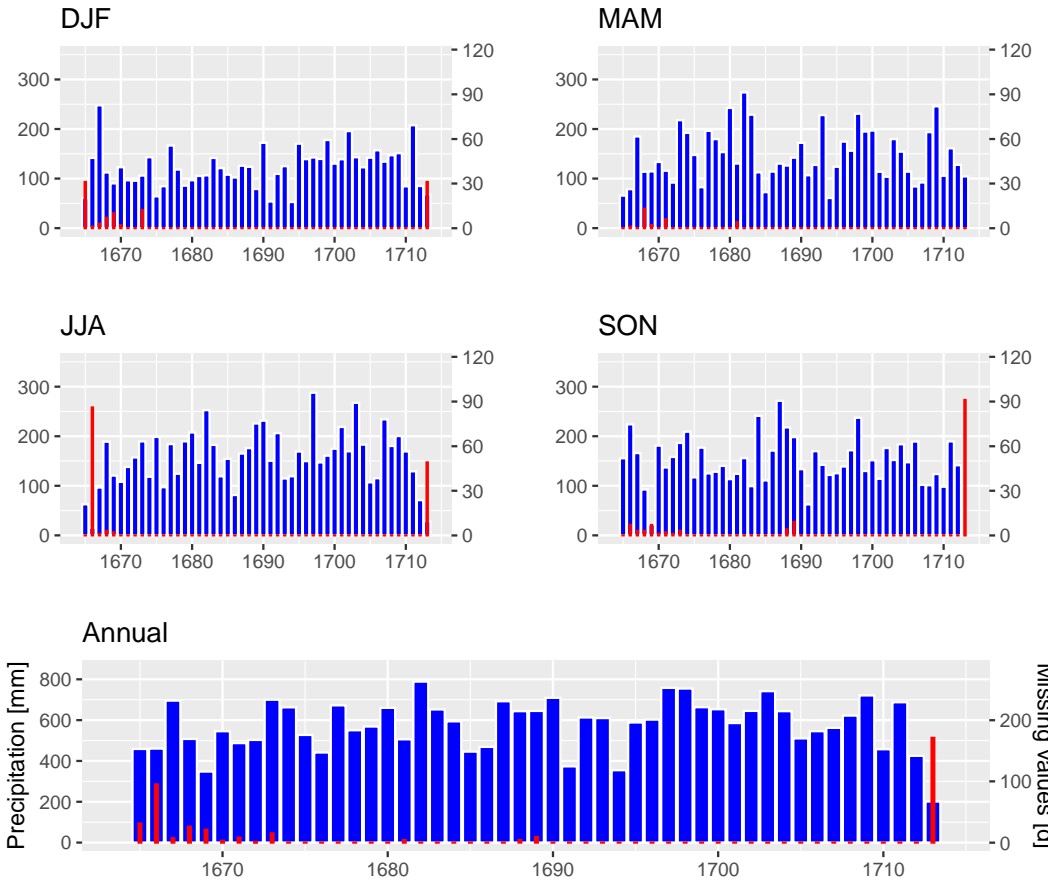

**Figure 10.** Seasonal (DJF, MAM, JJA, SON) and annual precipitation totals are shown as blue bars. The red bars show the days with missing entries in Morin's journal.



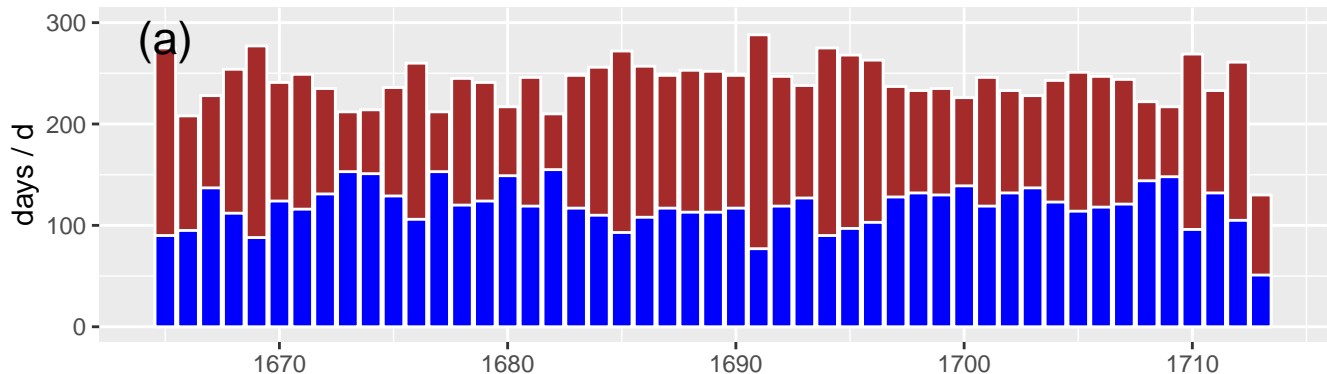

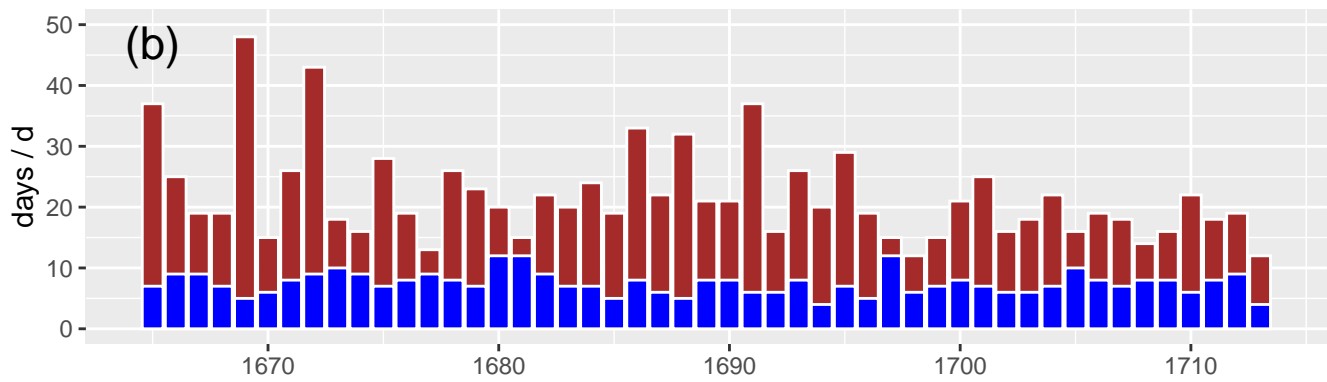

**Figure 11.** In (a), dry days (red bar; PT < 1 mm) and wet days (blue bar; PT >= 1 mm) are plotted. In (b), the contiguous dry days (red bar; CDD) and the contiguous wet days (blue bar; CWD) are plotted.

*Data availability.* All of the data used to perform the analysis in this study are described and properly referenced in the paper. The supplementary dataset can be found here: https://doi.org/10.5281/zenodo.7404635.





## A1  Appendix A figures

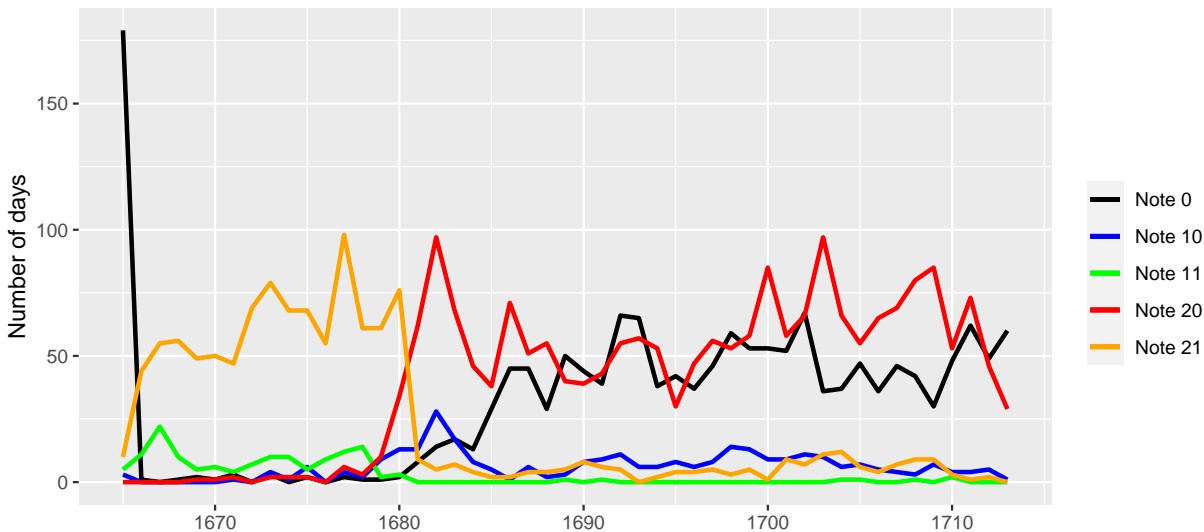

**Figure A1.** The colored lines show the time series of different notes in Louis Morin's journal of appearances per year





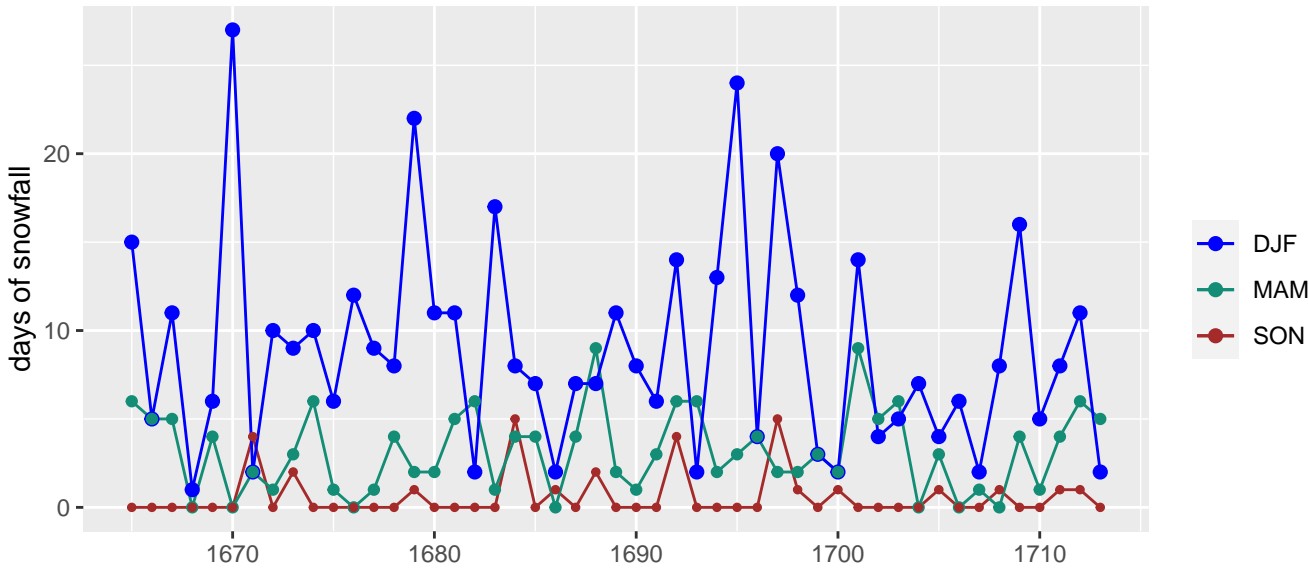

**Figure A2.** Total number of days when at least one note for snow was made per season (DJF, MAM, SON).



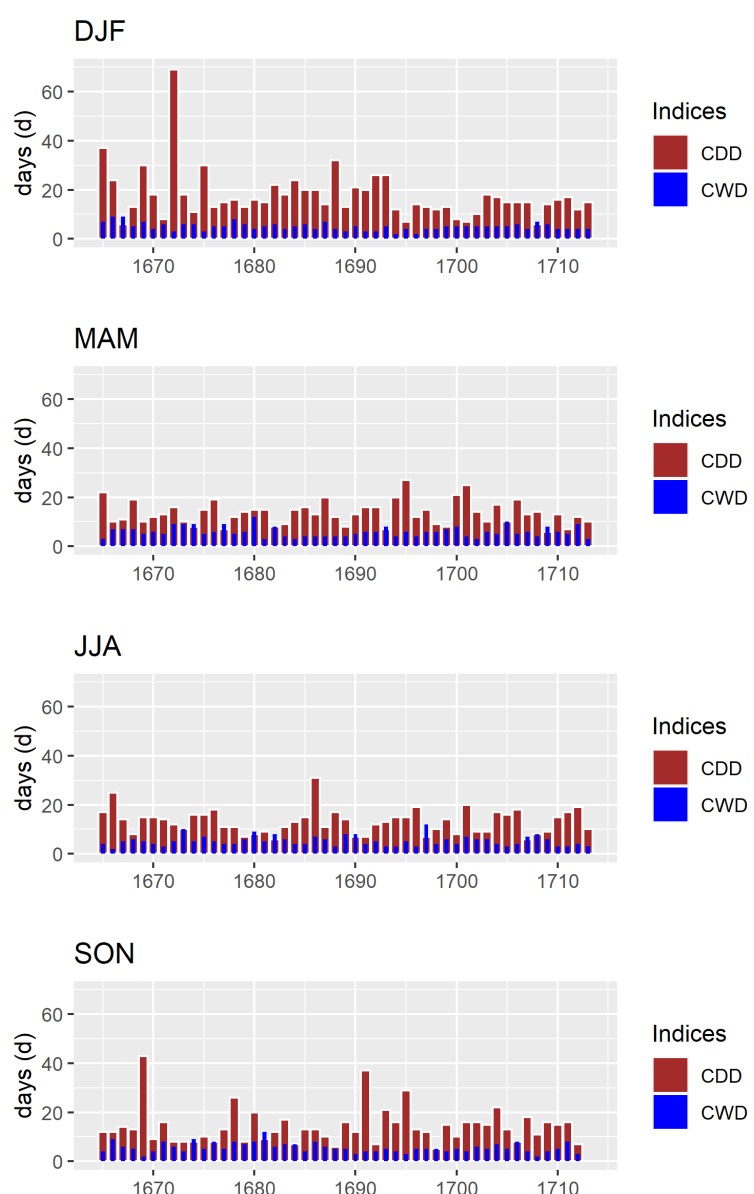

**Figure A3.** Consecutive dry days (CDD, red) and consecutive wet days (CWD, blue) for each season.



## A1  Appendix A tables

| a4 | b4 | c4 | d4 | RMSE | MAE |
|---|---|---|---|---|---|
| MJJAS | | | | | |
| 0 | 0.6 | 0 | 0.6 | 27.89 | 19.61 |
| 0 | 0.8 | 0.0 | 0.5 | 27.90 | 19.59 |
| 0 | 0.9 | 0.1 | 0.4 | 27.94 | 19.60 |
| 0.1 | 0.5 | 0.0 | 0.6 | 27.95 | 19.59 |
| 0.1 | 0.6 | 0.1 | 0.5 | 27.95 | 19.58 |
| 0.1 | 0.7 | 0.0 | 0.5 | 27.94 | 19.57 |
| 0.1 | 0.7 | 0.2 | 0.4 | 27.97 | 19.60 |
| 0.2 | 0.7 | 0.0 | 0.5 | 27.91 | 19.61 |
| 0.3 | 0.6 | 0.0 | 0.5 | 27.96 | 19.59 |
| ONDJFMA | | | | | |
| 0.0 | 0.2 | 0.0 | 0.5 | 19.16 | 13.38 |
| 0.0 | 0.3 | 0.1 | 0.4 | 19.26 | 13.33 |
| 0.0 | 0.4 | 0.0 | 0.4 | 19.27 | 13.26 |
| 0.1 | 0.1 | 0.0 | 0.5 | 19.18 | 13.29 |
| 0.1 | 0.2 | 0.1 | 0.4 | 19.35 | 13.38 |
| 0.1 | 0.4 | 0.0 | 0.4 | 19.27 | 13.29 |
| 0.2 | 0.2 | 0.1 | 0.4 | 19.31 | 13.38 |
| 0.2 | 0.3 | 0.0 | 0.4 | 19.31 | 13.29 |
| 0.3 | 0.0 | 0.0 | 0.5 | 19.23 | 13.35 |
| 0.3 | 0.3 | 0.0 | 0.4 | 19.32 | 13.35 |
| 0.4 | 0.2 | 0 | 0.4 | 19.35 | 13.34 |

**Table A1.** Parameters for Equ. 4 with the lowest Root Mean Square Error (RMSE) and Mean Absolute Error (MAE) for the warmer months (May to September) and colder months (October to April).





| Year | Jan | Feb | Mar | Apr | May | June | July | Aug | Sept | Oct | Nov | Dec | Ann |
|------|------|------|------|------|------|------|------|------|------|------|------|------|------|
| 1665 | 6.9 | 56.2 | 10.4 | 7.3 | 58.0 | 35.9 | 27.9 | 59.4 | 24.0 | 43.6 | 101.4 | 27.7 | 458.7 |
| 1666 | 39.7 | 46.8 | 19.8 | 50.4 | 8.4 | 16.2 | NaN | NaN | 78.6 | 68.9 | 76.5 | 55.2 | 460.5 |
| 1667 | 57.0 | 102.4 | 84.0 | 9.9 | 91.2 | 28.8 | 17.4 | 50.4 | 55.2 | 69.6 | 41.2 | 88.5 | 695.6 |
| 1668 | 17.8 | 30.8 | 13.0 | 55.7 | 45.0 | 55.2 | 58.8 | 75.0 | 12.0 | 21.2 | 59.4 | 63.8 | 507.7 |
| 1669 | 6.8 | 50.8 | 28.1 | 22.2 | 64.2 | 42.0 | 32.4 | 46.2 | 0.0 | 16.3 | 6.3 | 32.7 | 348.0 |
| 1670 | 33.4 | 46.6 | 29.9 | 52.0 | 51.6 | 16.8 | 38.4 | 52.8 | 78.0 | 64.9 | 38.2 | 42.9 | 545.5 |
| 1671 | 53.0 | 37.0 | 39.4 | 44.0 | 32.4 | 70.2 | 29.4 | 38.4 | 61.8 | 60.7 | 14.5 | 6.5 | 487.3 |
| 1672 | 0.5 | 26.3 | 28.9 | 32.4 | 30.6 | 18.6 | 71.4 | 67.2 | 64.2 | 52.0 | 42.0 | 68.8 | 502.9 |
| 1673 | 58.2 | 24.2 | 78.1 | 72.6 | 67.1 | 83.4 | 76.2 | 30.0 | 67.8 | 64.2 | 54.4 | 23.7 | 699.9 |
| 1674 | 45.3 | 63.6 | 83.2 | 34.6 | 74.4 | 19.8 | 58.8 | 39.6 | 104.4 | 66.4 | 38.3 | 34.2 | 662.6 |
| 1675 | 10.6 | 31.0 | 52.1 | 48.7 | 46.8 | 79.2 | 58.8 | 60.6 | 43.2 | 30.0 | 43.8 | 22.6 | 527.4 |
| 1676 | 25.4 | 22.2 | 27.0 | 24.4 | 31.2 | 49.2 | 22.8 | 25.2 | 94.2 | 56.5 | 26.2 | 37.0 | 441.3 |
| 1677 | 41.2 | 79.4 | 77.3 | 46.4 | 72.6 | 79.2 | 57.0 | 48.0 | 24.6 | 47.4 | 53.1 | 46.1 | 672.3 |
| 1678 | 34.6 | 44.1 | 86.5 | 27.1 | 66.0 | 77.4 | 24.0 | 22.8 | 26.4 | 84.8 | 17.1 | 39.5 | 550.3 |
| 1679 | 29.7 | 36.4 | 39.5 | 92.3 | 21.6 | 60.0 | 30.6 | 99.0 | 61.2 | 51.3 | 27.6 | 19.5 | 568.7 |
| 1680 | 42.7 | 26.3 | 54.4 | 104.1 | 84.0 | 101.4 | 57.0 | 49.2 | 4.8 | 32.2 | 76.4 | 27.6 | 660.1 |
| 1681 | 12.6 | 31.9 | 71.8 | 38.1 | 20.4 | 28.8 | 86.4 | 31.1 | 35.6 | 21.4 | 67.0 | 60.8 | 505.9 |
| 1682 | 62.3 | 9.9 | 95.4 | 68.0 | 110.4 | 75.6 | 95.4 | 81.6 | 68.2 | 55.7 | 31.9 | 34.3 | 788.7 |
| 1683 | 22.1 | 30.0 | 74.9 | 56.3 | 97.7 | 39.4 | 51.4 | 91.7 | 32.9 | 15.1 | 51.3 | 89.5 | 652.3 |
| 1684 | 23.3 | 51.1 | 24.5 | 33.1 | 55.0 | 12.0 | 51.5 | 55.7 | 96.5 | 67.1 | 77.5 | 46.7 | 594.0 |
| 1685 | 45.1 | 36.6 | 23.5 | 35.4 | 13.7 | 41.6 | 83.5 | 29.2 | 26.4 | 36.0 | 48.5 | 26.1 | 445.6 |
| 1686 | 57.7 | 5.8 | 31.2 | 17.2 | 65.7 | 35.9 | 8.5 | 37.0 | 45.7 | 49.0 | 76.4 | 38.7 | 468.8 |
| 1687 | 21.2 | 29.3 | 32.0 | 41.2 | 56.9 | 43.7 | 60.7 | 60.6 | 164.7 | 25.7 | 81.1 | 75.3 | 692.4 |
| 1688 | 47.3 | 2.4 | 26.8 | 48.8 | 50.7 | 89.3 | 53.3 | 33.4 | 69.3 | 55.4 | 93.1 | 74.0 | 643.8 |
| 1689 | 46.2 | 19.1 | 50.1 | 49.6 | 42.4 | 84.6 | 108.4 | 32.7 | 72.4 | 97.3 | 28.5 | 13.7 | 645.0 |
| 1690 | 102.8 | 52.8 | 35.5 | 25.7 | 110.9 | 84.1 | 49.5 | 97.8 | 37.4 | 69.1 | 27.0 | 16.5 | 709.1 |
| 1691 | 27.9 | 13.6 | 18.5 | 29.1 | 59.1 | 52.6 | 77.3 | 20.5 | 53.1 | 7.3 | 2.1 | 12.3 | 373.4 |
| 1692 | 20.7 | 47.7 | 35.0 | 31.2 | 61.3 | 51.6 | 115.7 | 39.0 | 99.3 | 49.4 | 21.1 | 41.0 | 613.0 |
| 1693 | 46.0 | 25.4 | 103.5 | 46.7 | 77.9 | 52.4 | 22.6 | 39.8 | 65.2 | 27.2 | 49.8 | 53.6 | 610.1 |
| 1694 | 9.8 | 13.1 | 11.8 | 8.9 | 40.1 | 47.9 | 26.2 | 45.1 | 32.1 | 20.5 | 69.3 | 29.8 | 354.6 |
| 1695 | 80.5 | 64.6 | 48.0 | 35.4 | 40.4 | 26.0 | 49.5 | 93.6 | 90.4 | 32.3 | 2.6 | 25.2 | 588.5 |
| 1696 | 30.0 | 30.8 | 26.1 | 22.8 | 125.7 | 83.5 | 32.8 | 33.4 | 43.0 | 50.2 | 45.7 | 78.4 | 602.4 |
| 1697 | 49.7 | 24.8 | 34.6 | 62.8 | 58.7 | 87.1 | 53.9 | 146.8 | 43.9 | 85.9 | 41.8 | 68.0 | 758.0 |
| 1698 | 37.5 | 47.2 | 33.0 | 66.1 | 131.7 | 78.2 | 45.8 | 23.2 | 90.4 | 72.1 | 74.7 | 54.9 | 754.8 |
| 1699 | 65.9 | 54.8 | 21.2 | 98.7 | 75.0 | 91.0 | 11.8 | 58.0 | 71.4 | 29.9 | 28.5 | 57.1 | 663.3 |

**Table A2.** Precipitation totals (mm) of each month and year from 1665–1713 (Part I).





| Year | Jan | Feb | Mar | Apr | May | June | July | Aug | Sept | Oct | Nov | Dec | Ann |
|------|-----|-----|-----|-----|-----|------|------|-----|------|-----|-----|-----|-----|
| 1700 | 28.6 | 57.3 | 60.9 | 83.2 | 52.8 | 83.5 | 63.6 | 27.6 | 15.9 | 53.7 | 81.9 | 44.2 | 653.2 |
| 1701 | 52.0 | 53.3 | 61.2 | 8.7 | 43.8 | 93.7 | 26.0 | 99.0 | 11.1 | 52.7 | 50.4 | 33.7 | 585.6 |
| 1702 | 63.8 | 41.7 | 35.5 | 50.3 | 17.8 | 27.8 | 67.5 | 74.3 | 56.6 | 44.7 | 74.9 | 90.3 | 645.2 |
| 1703 | 41.8 | 63.7 | 31.5 | 62.4 | 85.7 | 151.5 | 77.1 | 38.7 | 53.6 | 67.4 | 31.0 | 37.5 | 741.9 |
| 1704 | 38.0 | 41.4 | 55.4 | 47.6 | 51.3 | 82.0 | 17.0 | 84.0 | 101.5 | 21.8 | 60.5 | 43.3 | 643.8 |
| 1705 | 22.1 | 29.4 | 27.0 | 63.3 | 23.6 | 46.6 | 26.0 | 34.4 | 50.3 | 56.0 | 41.1 | 90.6 | 510.4 |
| 1706 | 32.1 | 63.5 | 19.8 | 11.2 | 53.1 | 52.3 | 37.3 | 25.7 | 69.2 | 40.0 | 80.1 | 61.5 | 545.8 |
| 1707 | 19.1 | 34.3 | 48.7 | 9.0 | 34.4 | 56.3 | 77.1 | 100.7 | 28.6 | 62.0 | 11.2 | 80.7 | 562.1 |
| 1708 | 79.7 | 35.8 | 57.9 | 55.7 | 80.1 | 69.3 | 66.8 | 44.2 | 53.7 | 33.2 | 14.2 | 31.6 | 622.2 |
| 1709 | 74.2 | 22.3 | 88.2 | 83.1 | 74.5 | 101.9 | 71.8 | 26.6 | 62.1 | 47.9 | 13.7 | 54.7 | 721.0 |
| 1710 | 29.3 | 15.9 | 34.2 | 41.5 | 29.9 | 30.6 | 60.4 | 78.3 | 30.6 | 22.4 | 45.1 | 39.1 | 457.3 |
| 1711 | 37.8 | 87.3 | 36.0 | 47.9 | 77.2 | 13.5 | 69.1 | 46.6 | 57.6 | 61.4 | 70.6 | 82.5 | 687.5 |
| 1712 | 39.2 | 26.3 | 16.2 | 96.1 | 15.3 | 19.6 | 36.0 | 15.4 | 40.7 | 47.7 | 53.0 | 19.5 | 425.0 |
| 1713 | 41.4 | 26.7 | 24.7 | 34.6 | 45.0 | 28.8 | - | - | - | - | - | - | 201.2 |

**Table A3.** Precipitation totals (mm) of each month and year from 1665–1713 (Part II).



*Author contributions.* Thomas Pliemon was responsible for conceptualization, data curation, formal analysis, investigation, methodology, resources, software, validation, visualization, and writing (original draft preparation, as well as review and editing). Ulrich Foelsche was responsible for conceptualization, funding acquisition, project administration, resources, supervision, validation, and writing (review and editing). Christian Rohr was responsible for resources, validation, and writing (review and editing). Christian Pfister was responsible for resources, validation, and writing (review and editing)

*Competing interests.*  The contact author has declared that neither they nor their co-authors have any competing interests.

*Acknowledgements.*  This research has been supported by the Austrian Science Fund (grant no. Clim_Hist_LIA P31088-N29). We thank PRF (www.proof-reading-service.com/en/) for the conscientious proofreading.



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
