# Peer review of "Precipitation reconstructions for Paris based on the observations by Louis Morin, 1665–1713 CE"

_EGUsphere, 2022_

## Referee Comment (RC2)

[referee-annotated manuscript omitted]

---

## Author Comment (AC1)

We thank Prof. Mock for the comments made on our study. In the following response, the italics indicate Prof. Mock's comments and the remaining text indicates our response letter.

*Overall, this is a very good paper that provides a unique approach in historical climatology on quantitative precipitation reconstruction for Paris in the Little Ice Age. Morin provided a unique record to conduct such a study. The authors provide nice descriptions of the data and methods (including the transfer functions), as well a potential drawback issues such as wind and evaporation on precipitation, comparisons with other types of data that include isotopes. The visuals mostly look quite good. I only have a few minor comments.*

*1) Around Line 65, Section 2.1. Did Morin also have any seasonal aspect on his observations times due to changes in daytime, sunlight, etc.? Also, with daytime observation times, I assume potential lack of rain observations at night? (briefly touched on Line 140 but should be expanded). If there is any nocturnal aspect to precipitation generating controls, those would be missing in the reconstruction.*

Morin's measurements do not appear to show significant seasonal differences. These would, if so, affect the meteorological variables that he normally recorded three times at fixed times of the day. Precipitation, however, was recorded as many as six times a day on rare occasions. This also means that precipitation records cannot be determined precisely in terms of time. He also seems to have recorded at night, as can be seen for example in Fig.1, where the note extends over both days. This note may have been made at 2 am, after he got up. Moreover, Fig. 7 compares the number of wet days of the Morin data with those of the E-OBS data. Here, only slight monthly deviations are recognizable, which for us do not require a correction of the reconstruction. However, we see the need to make the point (possible underestimation of precipitation during the night / sleeping hours) more extensive and clearer than we have done so far. The reconstruction itself we will keep as before, only we will point out possible errors in the continuous text more extensively.

*2) Figure 3 and around Line 190. I am pleased to see scatterplots for the calibration, but wonder with the Pearson's Correlation used, it utilizes the more non-parametric aspects of the precipitation data. Perhaps using Spearman's Correlation should also be included. The non-parametric aspects also apply elsewhere such as Figure 5.*

Thank you for this important point. We will add the Spearman Correlation coefficient to the figures.

*3) Line 19. Precipitation and temperature are not the only important climatic elements, probably the most straightforward elements.*

Thank you for this comment. We will slightly correct the meaning of the sentence.

*4) For some of the figures, all axis titles should be capitalized (first letter) for consistency, and the font size could be increased in some of the figures (ex. Figure 9) Figure 4 provides an ideal example.*

Thank you for this comment. We will correct the points raised.

---

## Author Comment (AC2)

We thank Prof. Macdonald for the positive assessment and the minor comments made on our study. In the following response, the italics indicate Prof. Macdonald's comments, and the remaining text indicates our response letter.

*This is a well written and detailed paper exploring the diaries and records of Louis Morin, Paris (1665-1713). The paper provides a detailed analysis, which is sound and robust, in places additional detail would enable the reader to develop a more detailed understanding of Morin and his intentions, but the analysis is good. Some minor improvements in clarity or expression may be achieved and an annotated copy of the manuscript is attached to help the authors undertake these changes. I have also flagged in a couple of places sections where I felt the arguments presented warranted reflection and were other works could be used to support arguments.*

We will consider all suggested minor revisions for improvement, which are explicitly stated in the attached file, in the revised paper.

*You might consider comparing the snow series information presented to that from the Manley snow series for London, a rare comparable record for the period.*

Thank you for this valuable advice. We will provide a comparison between the Manley snow series and that of Morin.

Please note that we make **additional adjustments**:

a) Change of the title to: "Precipitation reconstructions for Paris based on the observations by Louis Morin, 1665–1713 CE
b) Slightly different result for DJF in Fig. 10.
c) Slightly different result in Fig. A2.
d) Additional sentence in the Data section stating that we have been given access to the manuscript and copies of the manuscript by the University of Bern and the Institut de France.

Best regards,

The authors

---

## Author Response (AR2)

**List of revisions:**

Title: Changed the word "of" to "by".

Line 3 ff: Restructured the sentence.

Line 15 ff & Line 293 ff & Line 334 ff & Table 3: Due to a coding error, DJF values changed, and we revised these accordingly (see also Fig. 10 and Fig. A2).

Line 19: Changed the word "important" to "straightforward".

Line 24: Changed "This instrument" to "The rain gauge was".

Line 26 ff: Added a citation (Lundstad et al 2023)

Line 27: Restructured the sentence as suggested.

Line 34: Restructured the sentence as suggested.

Line 43: Added a sentence on a work of citizen science.

Line 53 f: Adopted the proposed changes.

Line 68: Adopted the proposed changes.

Line 77: Added the start and end dates of the observations.

Line 79 f: Restructured the sentences.

Line 82: Added more information on the life of Louis Morin.

Line 86: Added a short statement that Morin didn't leave any information, which instruments he used.

Line 86: Additional sentence stating that we got access to the manuscript and copies of the manuscript by the University of Bern and the Institut de France.

Line 86: Additional statement on the metadata of the measurement instruments.

Line 90 ff: Added two citations, especially one recently published paper dealing with Morin's humidity measurements.

Line 96: Changed the meaning of the sentence accordingly as suggested.

Line 98: Adopted the proposed changes.

Line 110 ff: Restructured the paragraph and extended the paragraph with possible uncertainties when dealing with precipitation reconstructions based on indexes.

Figure 3: Added the Spearman correlation coefficient.

Figure 5: Added the Spearman correlation coefficient.

Figure 10: Due to a coding error for the DJF season, we provide a new plot with slightly different results.

Figure A2: We provided a comparison between the Manley snow series and that of Morin in the text. Furthermore, due to a coding error for the DJF season, we provide a new plot with slightly different results.

Ad all figures: We have adjusted the axis labels.

Additional change, which was accepted by the editor via e-mail:

Section 2.1: Added that Morin was born in Le Mans and not in Paris.